# pH-Dependent Leaching Characteristics of Major and Toxic Elements from Red Mud

**DOI:** 10.3390/ijerph16112046

**Published:** 2019-06-10

**Authors:** Yulong Cui, Jiannan Chen, Yibo Zhang, Daoping Peng, Tao Huang, Chunwei Sun

**Affiliations:** 1Faculty of Geosciences and Environmental Engineering, Southwest Jiaotong University, Chengdu 611756, China; yulong_cui_swjtu@126.com (Y.C.); yibo@swjtu.edu.cn (Y.Z.); sunchunwei0310@gmail.com (C.S.); 2SWJTU-Leeds Joint School, Southwest Jiaotong University, Chengdu 611756, China; taohuang70@126.com

**Keywords:** pH, red mud, leaching, geochemical model, leaching control mechanism

## Abstract

This study analyzes the leaching behavior of elements from red mud (bauxite residue) at pH values ranging from 2 to 13. The leaching characteristics of metals and contaminated anions in five red mud samples produced by Bayer and combined processes were analyzed using the batch leaching technique following the US Environmental Protection Agency (USEPA) Method 1313. In addition, the geochemical model of MINTEQ 3.1 was used to identify the leaching mechanisms of metals. The results showed that Ca, Mg, and Ba follow the cationic leaching pattern. Al, As, and Cr show an amphoteric leaching pattern. The leaching of Cl^−^ is unaffected by the pH. The maximum leaching concentration of the proprietary elements occurs under extremely acidic conditions (pH = 2), except for As. The leaching concentration of F^−^ reaches 1.4–27.0 mg/L in natural pH conditions (i.e., no acid or base addition). At the same pH level, the leaching concentrations of Pb, As, Cr, and Cu are generally higher from red mud produced by the combined process than that those of red mud from the Bayer process. The leaching concentration of these elements is not strongly related to the total elemental concentration in the red mud. Geochemical modeling analysis indicates that the leaching of metal elements, including Al, Ca, Fe, Cr, Cu, Pb, Mg, Ba, and Mn, in red mud are controlled by solubility. The leaching of these elements depended on the dissolution/precipitation of their (hydr)oxides, carbonate, or sulfate solids.

## 1. Introduction

Red mud (bauxite residue) refers to the industrial solid waste associated with the process of alumina smelting from bauxite ore, which mainly consists of alumina, silicate, iron, and titanium oxides [1]. Approximately 0.6–2.5 tons of red mud are produced for every ton of alumina [2,3]. Approximately 150 million tons of red mud are produced annually worldwide, and the annual generation of red mud has grown since 2015 [4,5]. Currently, deep-sea dumping, landfilling, and impoundment are the most common methods for red mud management [6]. The groundwater or soil near red mud management facilities may be polluted by the leachate or sludge from the red mud [7]. 

The leaching behavior of red mud was evaluated by field and laboratory leaching tests to assess the potential risks of red mud leachate to human health and the environment [8,9]. Sun et al. [1] conducted a nationwide evaluation of the chemical compositions in leachates from red mud across China. They found that red mud leachate is hyperalkaline (pH > 12) and contains high concentrations of aluminum (Al, 118.3–1327.4 mg/L), chloride (Cl^−^, 511.4–6588.1 mg/L), fluoride (F^−^, 88.0–299.6 mg/L), sodium (Na, 1200.5–10,650.0 mg/L), nitrate (NO_3_^−^, 183.2–730.7 mg/L), and sulfate (SO_4_^2−^, 502.5–6593.0 mg/L). These elements exceed the recommended groundwater quality standards of China up to 6637 times. Sun et al. [1] also found that the minor and trace elements, including arsenic (As, 0.2–2.0 mg/L), chromium (Cr, 0.1–5.9 mg/L), cadmium (Cd, 12–172 μg/L), mercury (Hg, 275–599 μg/L), and selenium (Se, 525–1359 μg/L), have a concentration up to 272 times higher than the maximum contamination levels (MCLs) of groundwater quality standards of both China and the US Environmental Protection Agency (USEPA). Rubinos et al. [10] evaluated the leaching behavior of trace metals from red mud using batch leaching procedures, including the toxicity characteristics leaching procedure (TCLP) and sequential leaching tests. They suggested that the concentrations of Cr, copper (Cu), and nickel (Ni) exceed the regulation limits of TCLP, and the releases of Al and Cr are pH-dependent. Ghosh et al. [11] used sequential leaching tests to analyze the leaching patterns of major (Al and iron (Fe)) and trace metals (Cu and Cr) in fresh and sintered red mud. They found that approximately 10% of Al, 1% of Fe, 27.1% for Cu, and 9.0% of Cr were released during the sequential leaching tests, and sintering the red mud may enhance the leaching of Al and Fe, but may negatively impact the leaching of Cu and Cr. 

Red mud has been considered as an engineering material due to its clay-like structure and alkaline nature [12,13]. Studies have shown the effectiveness of replacing cement with red mud, which indicates that red mud may have a similar pozzolanic nature to that of cement or incineration ashes [14]. The leaching of elements, especially hazardous elements, is mainly dependent on the environmental conditions [15]. These conditions include pH, temperature, reaction time, liquid-to-solid ratio. Uzun et al. [16] evaluated the leaching of metal from red mud by increasing the agitation rate. They found the total dissolution increased from 5% to 23.18% by agitating up to 400 rpm. Rachel et al. [17] found acid addition (5 mol/L) and temperature (80 °C) can significantly enhance the metal extraction from red mud. Lim and Shon [18] found that acid concentration (6 N sulfuric acid), leaching temperature (70 °C), and reaction time (2 hours with ultrasound) could enhance the metal leaching from red mud, while the increase of the solid-to-liquid ratio (from 2% to 4%) reduces the metal leaching. Among these factors, pH is the most important parameter that controls the release of inorganic constituents from the solid phase [19,20]. The leaching behavior as a function of pH will help estimate the mobility of elements from red mud in various environmental conditions of geotechnical applications. However, the pH-dependent leaching mechanisms of elements from red mud have not yet been well understood.

This study aims to investigate the leaching characteristics of major and trace elements in red mud under different pH scenarios (from 2–13) using the batch leaching technique. The experimental data were imported into the geochemical modeling program MINTEQ (KTH, Stockholm, Sweden) 3.1 to determine the primary oxidation state of the elements, analyze the leaching mechanisms of the elements, and subsequently predict the elements’ leaching control mechanism.

## 2. Material and Methods

### 2.1. Sampling and Mineralogy of Red Mud

In this study, red mud samples were collected from five management facilities located in three provinces (i.e., Guangxi, Shandong, and Henan) of China. Red mud samples GX-A-B and GX-B-B were collected from different manufacturers in Pingguo County and Jingxi County located in Guangxi Province, respectively. Red mud samples SD-A-B and SD-B-B were collected from different manufacturers but in the same area in Zibo, Shandong. Red mud sample HN-A-C was collected in Xingyang, Henan Province. Fresh red mud samples (produced within 7 days), i.e., GX-A-B, GX-B-B, SD-A-B, and HN-A-C, were collected after the pressure filtration (before being filled into the red mud reservoir), and the dried red mud SD-B-B was directly sampled from the red mud reservoir. An initial 100 kg of each red mud was collected by a forklift and mixed uniformly by shovels. Then, 20 kg of red mud sample was collected in a sealed container and transported to the laboratory for leaching tests. Four red mud samples were obtained from the Bayer process. Sample HN-A-C went through the Bayer-sintering combined process. Quantitative X-ray diffraction analysis (Rigaku D/MAX-2005 X-ray diffractometer, Tokyo, Japan) was performed on the red mud samples to determine the most prevalent mineralogical compositions. The main mineral phases include quartz, calcite, hematite, hydrogarnet, sodalite, anhydrite, cancrinite, and gibbsite (Table 1).

### 2.2. Red Mud Physical Properties 

Table 2 summarizes the physical properties of the five red mud samples. The moisture content per red mud was analyzed following the procedure in American Society for Testing and Materials (ASTM) D2216. Fifty grams of each red mud was dried in an oven at a temperature of 110 ± 5 °C for 24 h to a constant mass. The moisture content was then calculated based on the masses of water and dry specimen. The moisture content of red mud ranged from 11.0% to 29.0%. The particle size distribution of red mud was tested by ASTM D2487. Each oven-dried (at 110 ± 5 °C) sample was screened by a series of standard sieves, including No. 4 (4.75 mm openings), No. 10 (2 mm), No.14 (1 mm), No. 35 (0.5 mm), No. 60 (0.25 mm), and No. 200 (0.075 mm), respectively. The particle size distribution was calculated by the weight of solid retained on each sieve. Loss on ignition (LOI) was performed by sintering samples at 900 °C using a muffle furnace, and the LOI results of the red mud samples ranged from 7.7% to 12.8%. According to the Unified Soil Classification System (USCS), red mud samples were mostly sandy or clayey materials. GX-B-B was classified as CL-ML (clay-silt with low plasticity), HN-A-C was classified as CH (clay with high plasticity), while GX-A-B, SD-A-B, and SD-B-B were classified as SC (clayey sand).

### 2.3. Total Elemental Compositions and Carbon Analysis

A total elemental composition analysis was conducted by acid digestion according to ASTM D5198-09, for which 5 g of the sample was digested at 90–95 °C for 2 hours with a 1:1 nitric acid digestion. The elemental compositions after digestion were measured via inductively coupled plasma-optical emission spectrometry (ICP-OES, Vista-MPX CCD Simultaneous ICP-OES, Varian Inc., CA, USA) and inductively coupled plasma-mass spectrometry (ICP-MS, Agilent 7700x ICP-MS, Agilent Technologies Inc., CA, USA). A carbon analyzer (SC144 DR LECO Inc., St. Joseph, MO, USA) was used to determine the total carbon (TC), total inorganic carbon (TIC), and total organic carbon (TOC). 

Table 3 summarizes the total elemental compositions and carbon content in the red mud samples. Concentrations of major elements (mass ratio >1% wt./wt.) are reported in mass percentage (%wt./wt.), while concentrations of trace elements (<1%) are reported in µg/g. As (28.7–203 µg/g), Pb (43.7–132 µg/g), and Cr (480–1370 µg/g) were the most abundant trace elements in the samples. In addition, the total carbon contents in the combined process red mud (i.e., HN-A-C, TC = 1.6%) were higher than the red mud from the Bayer process (TC = 0.7–1.3%), which may result from the addition of the lime in limestone during the combined process [1].

### 2.4. pH-Dependent Leaching Analysis

pH-dependent batch leaching tests were conducted following USEPA method 1313. Samples at a liquid-to-solid ratio of 10:1 by weight were agitated in an end-over-end tumbler at a speed of 30 r/min for 24 h. A preliminary test on the effect of contact time (0–72 h) on the leaching experiment indicated 24 h is sufficient for each batch to reach chemical equilibrium condition (pH, EC, and elements reached constant). A pH range of 2–13 was used with target pH values of 13.0, 12.0, 10.5, 9.0, 8.0, 7.0, 5.5, 4, and 2.0, respectively. A pretest titration was conducted to determine the equilibrium time and the acid/base addition required for each target pH value, while the electrical conductivity (EC) and oxidation-reduction potential (ORP) value were determined after testing. An acid neutralization capacity (ANC) curve was also obtained from the certain acid/base addition and corresponding pH value reading. Triplicate tests were carried out for each sample. Leach XS model recommended by the USEPA 1313 method was used for an acid/base addition calculation for each pH target [24]. The leaching eluate that reached the target pH value was filtered through a 0.45 μm membrane disk filter using a 20 ml syringe into a 15 mL high-density polyethylene tube bottle and stored at 4 °C for subsequent analysis. 

The pH value and ORP of the eluate were determined via pH meter (PHS-3E pH meter, SPSIC Ltd., Shanghai, China). The leaching concentrations of the elements in the eluate, including major elements (Na, Al, Ca, Fe, Ti, Si), trace elements (K, Mg, Co, Li, Mo, Ni, Zn), and heavy metals (As, Ba, Cr, Cu, Pb), were analyzed by ICP-OES and ICP-MS. The anionic components of chloride (Cl^−^), fluoride (F^−^), sulfate (SO_4_^2−^), and Nitrite (NO_3_^−^) in the eluate were measured via an ion chromatograph (IC, ICS-1100, DIONEX Inc., CA, US).

### 2.5. Geochemical Modeling Analysis

A geochemical equilibrium model is a popular tool used to predict the ionic phases of elements and the saturation index of minerals in aqueous solutions at equilibrium [25]. Geochemical modeling was used in the current study to investigate the solubility control mechanism of elements. The numerical model Visual MINTEQ (ver. 3.1) developed by USEPA was used to identify the predominant oxidation states and leaching mechanism of the metals [26]. 

A two-step modeling process was conducted with MINTEQ. In the first step, an equilibrium between the leaching eluate and atmospheric temperature was assumed at 25 °C, since the leaching and filtration processes were exposed to the atmosphere. The dominant oxidation states of the metals that were estimated from MINTEQA 3.1 predicted the aqueous concentrations of the species. In the second step, the solid-liquid phase saturation index (SI) in the eluate at a fixed pH was calculated in the liquid phase by introducing complexation reactions [27]. The input of the modeling included leachate pH, elemental concentrations, and temperature. The simulation selected the main oxidized components: Al^3+^, Ca^2+^, Fe^3+^, Mg^2+^, Ba^2+^, Cu^2+^, Pb^2+^, Cr (Ш) as Cr (OH)^2−^, SO4^2−^, and CO_3_^2−^. The gas phase condition is assumed to be the atmospheric partial pressure of CO_2_ (3.16 × 104 atm, or 32.02 N/m^2^) [28]. The geochemical model MINTEQ3.1 presented the simulation results by calculating the logarithm of the activity of the substance, and the single ion activity coefficient was calculated by the Davies equation. The pH-log activity of the leachate was combined to determine whether the simulated element leaching behavior was controlled by mineral solubility [29]. If the leaching of an element is controlled by the solubility of a specific mineral, the log activity will fall near the solubility/stability line of the mineral. 

## 3. Results

### 3.1. pH-Dependent Leaching Tests

#### 3.1.1. Acid Neutralization Capacity 

The ANC curves of the five red mud samples are shown in Figure 1, where the negative value and the positive value of the X-axis represent the base and acid addition amounts, respectively. The ANC reflects the buffering of the red mud to acid attack, which affects the rate and degree of contaminant leaching [30]. The natural pH (the original pH of samples with no acid or base added) of the red mud samples ranged from 10.5 to 11.0, with GX-A-B and HN-A-C having relatively higher natural pH values (≈11.0). Generally, all five samples showed a similar acid neutralization trend, where a rapid drop of pH existed right after the acid was added, and a plateau occurred at approximately pH = 6. Garrabrants et al. [31] found that the pH plateau at 2 < pH < 6 of ANC is generally due to the dissolution of CaCO_3_ in cement-based material. In this study, calcite was detected in the mineral phase of the red mud samples, which is generally produced by the carbonation of mineral components (such as hydrous hydrate and calcium silicate hydrate) after processing or during the stockpile period [31]. Additionally, the result of TIC (Table 2) indicates that GX-A-B (1.1% by mass) and HN-A-C (1.0% by mass) have a stronger acid buffering capacity (more acid added at pH 2–6) due to the higher inorganic carbon content than other samples (TIC of GX-B-B, SD-B-B, and SD-A-B are 0.9%, 0,7%, and 0.4%, respectively). 

#### 3.1.2. Oxidation-Reduction Potential

Figure 2 shows the eluate ORP as the function of pH in the pH-dependent leaching tests. The oxidation-reduction potential is a widely used parameter for characterizing chemical or biological redox processes. ORP is an indicator of the oxidation or reduction environment (ORP > 0: oxidizing environment, ORP < 0: reducing environment) [32]. The ORP values show an almost linear negative correlation with pH and become negative (indicating a reducing environment) in alkaline conditions (pH > 11.5). No substantial difference in the ORP tendency between the combined and Bayer red muds was found. Generally, solutions with a higher pH have more oxidized metal element groups, while solutions with low pH contain relatively more dissociated metal ions [33].

#### 3.1.3. Leaching of Major Elements

The pH-dependent leaching behavior of elements generally follows three distinct patterns: cationic, oxyanionic, and amphoteric [34]. In the cationic pattern, the leachate concentrations of elements showed a consistent negative correlation with the pH, while the oxyanionic pattern showed the opposite trend. In the amphoteric pattern, the leachate concentrations of elements decreased first and then increased as the pH increased, presenting a V-shape. The leaching behaviors of major elements, i.e., Al, Ca, Fe, and Si, as a function of pH from the red mud samples are shown in Figure 3.

The release of Al from the red mud samples as a function of pH followed an amphoteric leaching pattern (Figure 3a), which is similar to results reported in previous studies [34,35,36]. The minimum leaching concentration was approximately 0.1–0.5 mg/L at pH = 7–8, and the maximum concentration reached 7.1 × 10^3^–9.4 × 10^3^ mg/L at pH = 2. Cama et al. [37] found that aluminosilicate phases are less stable compared to boehmite and gibbsite under acid attack. The concentration of Ca showed a continuous negative correlation with pH, thus representing the cationic pattern (Figure 3b). The decrease in pH induced strong acid attack on the Ca-bearing minerals and thus released a higher concentration of Ca in the eluate [38]. Zhang et al. [36] also reported the cationic leaching pattern of Ca from municipal solid waste incineration (MSWI) fly ash. The release of Ca from the combined process red mud was slightly higher than the Bayer process in acidic conditions. Two orders of magnitude difference in the Ca concentration (a maximum of 7.7 × 10^2^ mg/L from the Bayer red mud, and a maximum 1.2 × 10^4^ mg/L from the combined process red mud) was observed at pH = 2. The minimum concentration was 1.0 mg/L at pH = 13. 

Fe and Si present a similar leaching pattern (Figure 3c,d). Maximum releases of Fe and Si were observed at pH = 2, and the concentrations were 798-965 mg/L and 5196–7179 mg/L, respectively. The leaching concentration of Fe declines with the increase of pH at pH < 8. When the pH reached 9, the leaching concentration started to increase until the material pH (i.e., pH ≈ 11) was reached. The leaching concentration then decreased again at high pH (> 12) levels. The solubility of Fe is often controlled by the oxide minerals, e.g., hematite (Fe_2_O_3_), which may release Fe at both acidic and alkaline conditions [39]. However, when the addition of hydroxyl was applied, Fe tended to precipitate as hydroxide Fe(OH)_3_, which decreased the Fe concentration in the eluate (pH > 10.5–11). The leaching of Si was majorly due to the dissolution of silicates rather than quartz under acid attack [40]. In the alkaline condition, Ning et al. [41] also found the Si solubility to rise rapidly with a pH increasing from 9 to 10.6, where the formation of H_2_SiO_4_^2−^ and H_3_SiO_4_^−^ is considered to be [42]. 

#### 3.1.4. Leaching of Trace Elements

The pH-dependent leaching behaviors of trace elements in the eluate from the red mud samples are shown in Figure 4. The leaching behavior of As and Cr followed the amphoteric pattern (Figure 4a,b), and Cu and Pb did not exhibit a distinct amphoteric leaching pattern (Figure 4c,d). The maximum leaching concentrations of Cr (0.5–87.2 mg/L), Cu (0.3–5.5 mg/L), and Pb (2.0–4.4 mg/L) were found in acidic conditions (pH = 2), whereas the maximum release of As (0.2–4.2 mg/L) occurred in either acidic condition or alkaline conditions. The minimum leaching concentrations of Cu and Pb occurred between pH = 7–9 and pH = 5.5–8, respectively, and the concentrations were close to the MDL (0.01 mg/L). The minimum leaching concentration of As and Cr appeared at pH = 5.5, with the concentration close to the detection limit (As = 0.1 mg/L, Cr = 0.01 mg/L). The leaching concentrations of As and Cr then increased until pH = 13. Amphoteric elements have relative low released concentrations at pH = 5.5–6.5 due to the formation of relatively insoluble hydroxides [43,44], while, at both high pH and low pH conditions, As and Cr form oxyanion (i.e., AsO_4_^3−^, HAsO_4_^2−^, H_2_AsO_4_^−^, H_3_AsO_4_, and CrO_4_^2−^) or cation (i.e., Cr^3+^), respectively, which are soluble in the eluate [45,46]. Similar leaching behaviors of Pb, As, Cr, and Cu from lead and copper smelter slags were reported by Nabajyoti et al. [47] and Jarošíkováet al. [48]. Additionally, no differences in the leaching patterns of trace elements were evident between the red mud samples from the Bayer process and combined process. 

The leaching of Mg follows a cationic pattern (Figure 4e). The maximum leaching concentration (20.0–279.0 mg/L) of Mg was found at pH = 2. The maximum leaching concentration is directly related to the total Mg contents in the red mud samples (Table 2). The minimum leaching concentration of Mg was found at pH > 9. Previous studies indicated that the leaching of Ba and Mn are sensitive to the pH of the environment [49,50]. The leaching behavior of Ba in this study followed the cationic pattern with a maximum leaching concentration of 0.5–1.5 mg/L at pH = 2 (Figure 4f). From neutral (pH = 7) to alkaline conditions, the release of Ba is low (~0.02 mg/L) and remained constant. Like Ca, leaching of Mg and Ba tended to increase with decreasing pH due to competition with the hydrogen ion. Additionally, Astrup et al. [51] found that leaching of Ba in the eluate is associated with the solubility of Ba(S, Cr)O_4_. The leaching of Mn was independent of pH at pH > 8 (Figure 4g). At pH = 2, the maximum release was observed (1.3–16.7 mg/L) for Mn, and the concentration decreased with increasing pH until pH = 8, where Mn^2+^ cation tended to precipitate as Mn(OH)_2_ [52]. 

#### 3.1.5. Leaching of Toxic Anions

The leaching behaviors of F^−^ and Cl^−^ as a function of pH are shown in Figure 5. F^−^ leaching was greatly affected by the eluate pH. With the addition of acid, the leaching of F^−^ increased rapidly and stabilized after pH = 3. Meanwhile, when the pH increased from 7 to 13, a linear increase in the F^−^ concentration was observed (Figure 5a). The minimum leaching concentrations occurred in neutral and weak acidic conditions (pH = 5.5–7.0). Gong et al. [53] claimed that at a weakly acidic pH (5.5–6.5), fluoride is favorable to adsorb on Al_2_O_3_. However, the leaching behavior of Cl^−^ was more likely independent of pH (Figure 5b).

### 3.2. Geochemical Modeling with MINTEQ

#### 3.2.1. Leaching Mechanisms of Major Elements 

The log activities of Al^3+^, Ca^2+^, and Fe^3+^ in the red mud eluate at different pH values are shown in Figure 6. The Al concentration at pH values between 4 and 13 was mainly controlled by the dissolution/precipitation of aluminum hydroxide compounds, including amorphous Al(OH)_3_, Gibbsite (crystalline Al(OH)_3_) and Boehmite [γ-AlO(OH)] (Figure 6a). The leaching concertation of Ca from red mud was more controlled by calcium sulfates (gypsum (CaSO_4_.2H_2_O) and anhydrite (CaSO_4_)) at the pH range of 2–11. At higher pH values (pH > 11), calcium carbonates (calcite (CaCO_3_) and aragonite (CaCO_3_)) began to control the leaching of Ca (Figure 6b). The geochemical modeling indicated that leaching of Fe^3+^ was mostly controlled by the Fe-hydroxides, e.g., ferrihydrite (Fe(OH)_3_) and goethite (α-FeO(OH)) (Figure 6c).

#### 3.2.2. Leaching Mechanism of Trace Elements 

The log activity diagrams of trace elements Cr, Cu, Pb, Mg, Ba, and Mn as a function of pH are shown in Figure 7. The leaching of Cr and Pb from the red mud samples are mainly controlled by oxide or/and hydroxide, respectively. At pH values of 6–13, amorphous chromic hydroxide (Cr(OH)_3_ am), crystalline chromic hydroxide Cr(OH)_3_, and chromic oxide (Cr_2_O_3_) controlled the leaching of Cr from red mud (Figure 7a). At 9 > pH > 6, the leaching concentration of Pb was close to the MDL (0.01 mg/L, Figure 4d), which illustrates that Pb^2+^ tends to form Pb(OH)_2_ precipitation (Figure 7c). In alkaline conditions (pH > 8), tenorite (CuO) and malachite (Cu_2_(OH)_2_CO_3_) controlled the leaching of Cu from red mud (Figure 7b). At pH 8–13, dolomite (MgCa(CO_3_)_2_) and magnesite (MgCO_3_) controlled the leaching of Mg in a similar dissolution form. In the range of pH < 8, the log activity of Mg^2+^ moved away from the mineral stability line, which indicated that Mg existed in the soluble forms (Figure 7d). At pH 2 to 10, Ba leaching was mainly controlled by barite (BaSO_4_) minerals, while at higher p conditions, witherite (BaCO_3_) began to control the leaching of Ba (Figure 7e). The calculated log activity of Mn is close to the Mn(OH)_2 (s)_ line, thus indicating that the leaching of Mn was controlled by the dissolution/precipitation of Mn(OH)_2 (s)_ (Figure 7f).

## 4. Discussion

Generally, this study showed that the leaching behaviors of Al, Cr, and As follow a prominent amphoteric leaching pattern, while Cu and Pb do not follow a distinct amphoteric leaching pattern. Ca, Mg, Ba, Mn, Fe, and Si follow a cationic leaching pattern with the highest level of leaching concentration occurring in acidic conditions (pH~2). However, the leaching of Cl^−^ is not strongly related to the eluate pH. Among these elements, heavy metals such as Pb, As, Cr, and Cu are independent of the total elemental contents in the red mud since no distinct differences were observed in the eluate concentrations of these elements. This phenomenon confirms the solubility controlling mechanism of these elements. The contents of Pb, As, Cr, and Cu in red mud from the Bayer process (especially the GX-A-B, GX-B-B samples) are higher than those of the combined red mud (i.e., HN-A-C). 

The leaching rate (η) of heavy metals was calculated in this study, which is defined as the ratio of elemental contents released into the eluate to the total elemental contents in the red mud sample (Equation (1)): η = ρV/(mw)(1)
where ρ is the element concentration in the eluate, mg/L; V is the total eluate volume, L; m is the total amount of dry red mud in the leaching test, kg; and w is the total element content in the dry red mud, mg/kg. The calculated maximum leaching rates of Pb, As, and Cr are 54.5%, 33.7%, and 50.9%, respectively, and these rates are all from the HN-A-C sample. The maximum leaching rate of Cu can reach up to 57.9% (GX-A-B). 

F^−^ has an extremely high concentration in red mud leachate with potential environmental hazards [10,54,55]. The concentration of F^−^ in the eluate is generally approximately 1–100 mg/L in both alkaline (pH > 12) and acidic conditions (pH < 2). The concentration of F^−^ reached the order of 10^1^ mg/L even under natural pH conditions, thereby indicating that F^−^ should be noted as a potential environmental hazard.

Most of the investigated major elements have a solubility controlling mechanism. Mullite (Al_6_Si_2_O_13_) was reported to be the primary source of Al^3+^ in the formation of aluminum hydroxide compounds [56]. However, mullite is generally unstable, and no data on mullite exists in the MINTEQ 3.1 database. Therefore, further studies on Al leaching should investigate the solubility controlling mechanism of mullite. Zhang et al. [36] stated the leaching mechanism of Ca from MSWI fly ash was controlled by calcium sulfates (gypsum (CaSO_4_.2H_2_O) and anhydrite (CaSO_4_)), while in more alkaline condition (pH > 12), calcium carbonates (calcite (CaCO_3_) played a more critical role, also in this study. Garavaglia and Caramuscio [29] reported that the leaching of Fe^3+^ from coal fly ashes is controlled by ferrihydrite (Fe(OH)_3_), which is the same as the finding of the current study. 

The leaching of trace elements, excepting As, is controlled by the solubility of minerals. Cu is a sensitive element for redox reactions [26]. Murarka et al. [57] and Dijkstra et al. [58] also found that tenorite (CuO) and malachite (Cu_2_(OH)_2_CO_3_) controlled Cu leaching in coal-combustion residues and MSW bottom ash tests. These two minerals also controlled the Cu leaching from the red mud. Similar to Cu, Cr is a redox-sensitive element with its original form in red mud as Cr_2_O_3_ [29,46]. The hydroxide Cr controls the solubility of the leached Cr concentration at a pH range of 6–13 for red mud tested in this study. Bektas [59] et al. found that when pH > 6, Pb precipitates in the form of Pb(OH)_2_(s), which is similar to the red mud in this study. The controlling minerals for the Mg release in this study were dolomite (MgCa(CO_3_)_2_) and magnesite (MgCO_3_), which are similar to those reported in MSWI bottom ash by Dijkstra et al. [60]. Mudd et al. [19] and Fruchter et al. [20] claimed that barite (BaSO_4_) and carbonate compounds (i.e., witherite (BaCO_3_)) are likely formed in aqueous solutions of fly ashes, which could also be the controlling mineral for Ba from red mud. The leaching pattern of Mn in red mud is similar to the coal fly ash reported by Komonweeraket [34] et al. and Gitari [52] et al. They found a positive correlation between the Mn(OH)_2_ precipitation rate and the pH value.

## 5. Conclusions

In this study, pH-dependent leaching tests were conducted to investigate the leaching characteristics of major elements and trace elements at different pH levels. Leaching controlling mechanisms of metal elements were studied via the geochemical modeling program MINTEQ 3.1. Based on the findings of this study, the following conclusions and recommendations are drawn: 

Acid neutralization curves for the red mud samples show a pH plateau at 2 < pH < 6, which is due to the buffering of carbonates in the red mud. The ORP value negatively correlates with the pH. At pH= ~12, the ORP decreased to zero, indicating a reducing environment in the eluate.

The maximum leaching concentration of the metal elements occurred at pH = 2, except for As, Ca, Mg, and Ba, which followed the cationic leaching pattern. Al, As, and Cr showed an amphoteric leaching pattern. Leaching of Cl^−^ was affected by the pH conditions. There was no substantial difference in the leaching trend of elements from red mud produced by the Bayer or combined process. The combined process red mud had a higher leaching rate of Pb, As, Cr and Cu than the Bayer process. The leaching rates of these elements were independent of the total elemental content in the red mud samples.

Geochemical modeling analysis indicated that the leaching of metal elements, including Al, Ca, Fe, Cr, Cu, Pb, Mg, Ba, and Mn, in red mud are solubility controlled. The leaching of these elements depends on the dissolution/precipitation of their oxides, hydroxides, carbonate, and sulfate solids. The leaching controlling mechanisms of the two kinds of red mud samples were in accordance. 

## Figures and Tables

**Figure 1 ijerph-16-02046-f001:**
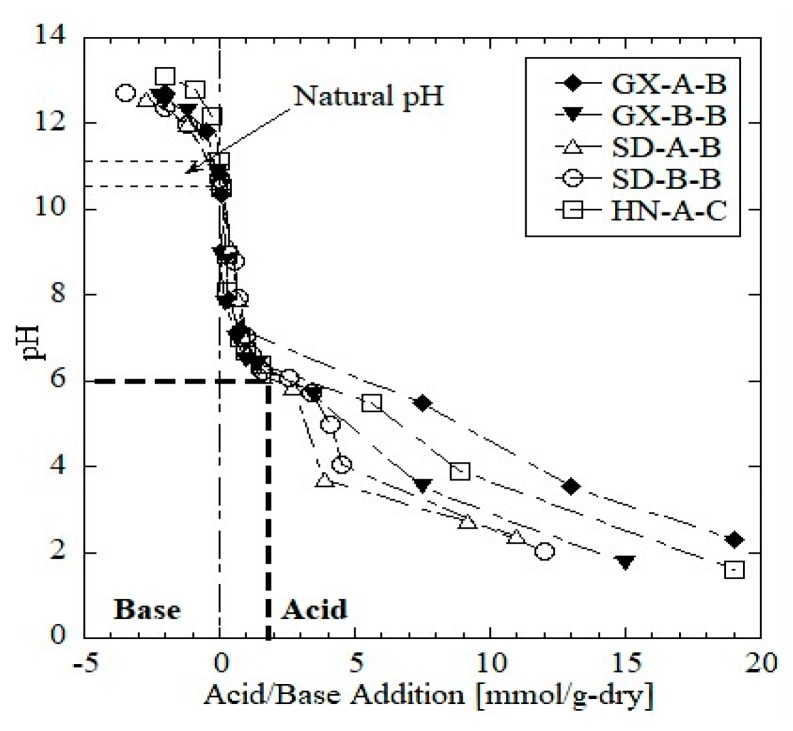
Acid neutralization capacity curves of red mud samples.

**Figure 2 ijerph-16-02046-f002:**
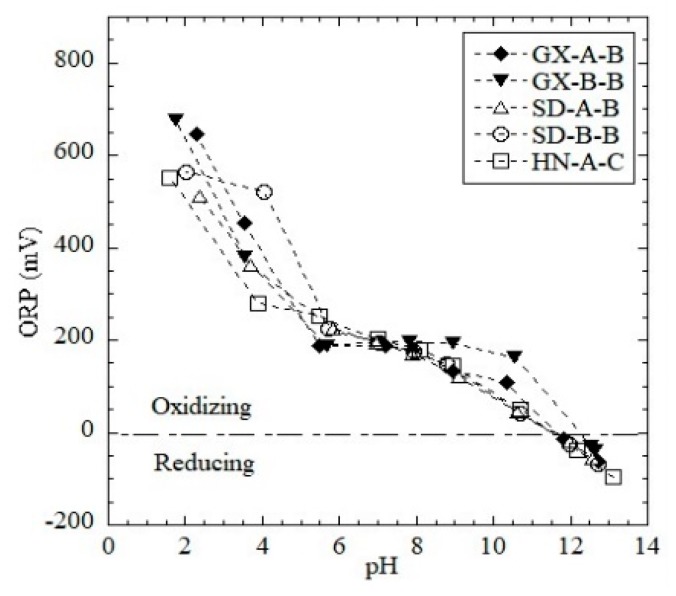
Oxidation-reduction potential as a function of pH in the eluate of pH-dependent leaching tests.

**Figure 3 ijerph-16-02046-f003:**
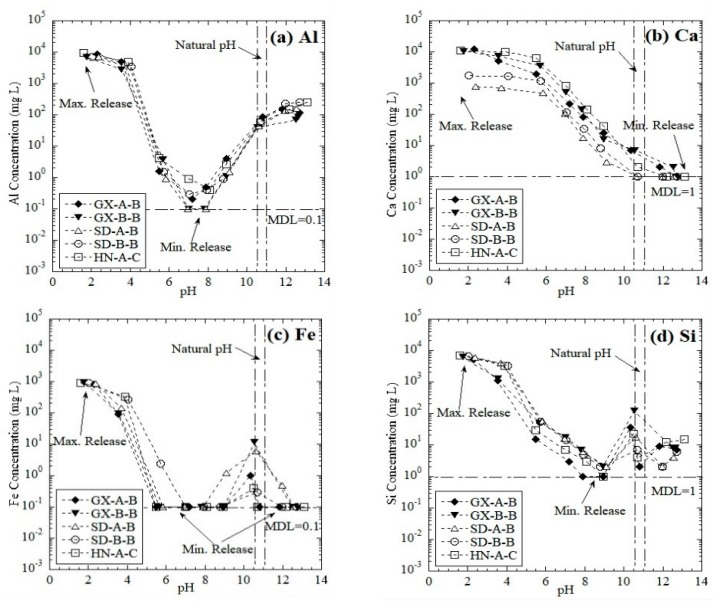
Leaching of major elements as a function of pH from red mud: (**a**) Al, (**b**) Ca, (**c**) Fe, and (**d**) Si. MDL = method detection limit.

**Figure 4 ijerph-16-02046-f004:**
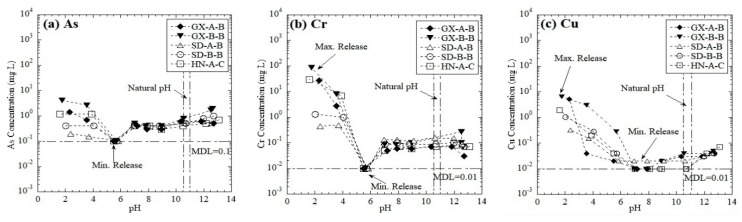
Leaching of trace elements as a function of pH from red mud: (**a**) As, (**b**) Cr, (**c**) Cu, (**d**) Pb, (**e**) Mg, (**f**) Ba, and (**g**) Mn. (MDL means method detection limit).

**Figure 5 ijerph-16-02046-f005:**
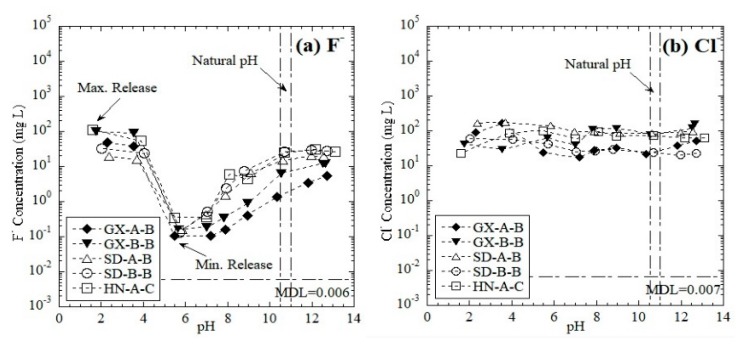
Leaching of toxic anions as a function of pH from RM: (**a**) F^−^ and (**b**) Cl^−^. MDL = method detection limit.

**Figure 6 ijerph-16-02046-f006:**
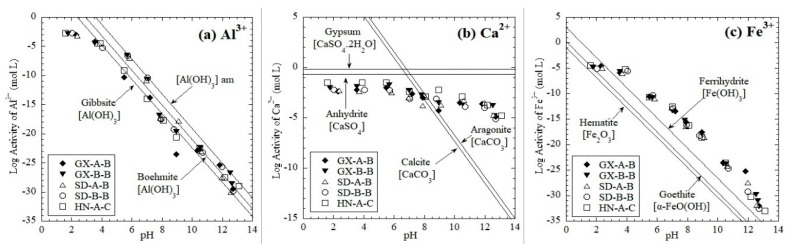
Log activity of major elements as a function of pH graphs: (**a**) Al^3+^, (**b**) Ca^2+^, and (**c**) Fe^3+^.

**Figure 7 ijerph-16-02046-f007:**
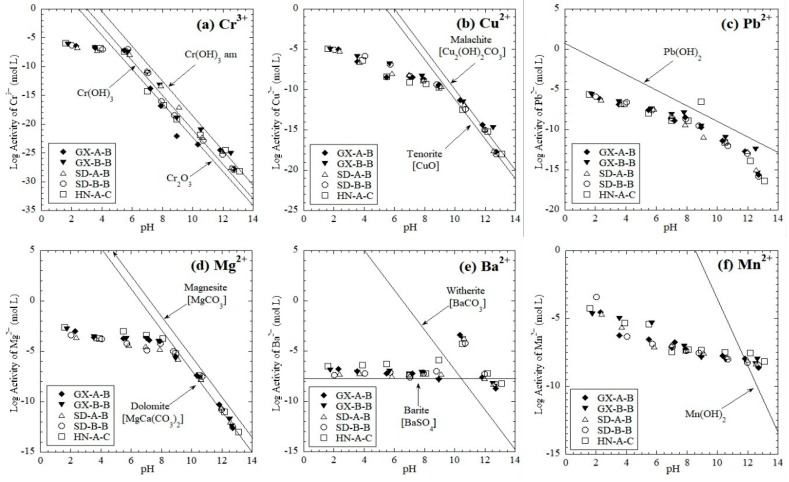
Log activity of trace metals as a function of pH (**a**) Cr^3+^, (**b**) Cu^2+^, (**c**) Pb^2+^, (**d**) Mg^2+^, (e) Ba^2+^, and (**f**) Mn^2+^.

**Table 1 ijerph-16-02046-t001:** Mineral phases (wt.%) of red mud samples in this study.

Mineral	Formula	GX-A-B	GX-B-B	SD-A-B	SD-B-B	HN-A-C
Quartz	SiO_2_	10	10		-	10
Calcite	CaCO_3_	-	20	10	10	-
Hematite	Fe_2_O_3_	20	40	35	35	50
Hydrogarnet	Ca_3_Al_2_(SiO_4_)_2_(OH)_4_	40	-	30	30	-
Sodalite	Na_8_(Al_6_Si_6_O_24_)Cl_2_	30	20	-	25	-
Anhydrite	CaSO_4_	-	10	-	-	-
Cancrinite	Na_6_Ca_2_((CO_3_)_2_Al_6_Si_6_O_24_)·2H_2_O	-	-	25	-	-
Gibbsite	Al(OH)_3_	-	-	-	-	40
		Bayer	Bayer	Bayer	Bayer	Combined

**Table 2 ijerph-16-02046-t002:** Physical properties of red mud sample used in this study.

Red Mud	Moisture Content (%)	Particle Size Distribution (mm) (%)					LOI (%)
		1–2	0.5–1	0.25–0.5	0.075–0.25	< 0.075	
**Method**	ASTM D2216	ASTM D2487					ASTM D7348
GX-A-B	20.2	7.1	19.2	14.0	15.1	44.6	11.6
GX-B-B	11.7	5.5	12.7	11.3	14.7	55.8	9.1
SD-A-B	29.0	4.7	16.0	15.0	17.5	46.8	7.7
SD-B-B	11.0	6.3	23.3	19.1	18.7	32.6	10.3
HN-A-C	24.1	1.0	5.2	14.2	23.5	56.1	12.8

**Table 3 ijerph-16-02046-t003:** Solid-phase concentration from total elemental analysis and carbon content.

Chemical Properties	Red mud samples					Ref.
	GX-A-B	GX-B-B	SD-A-B	SD-B-B	HN-A-C	
Major elements (%)	ASTM D5198					
Aluminum (Al)	9.1	8.3	9.2	9.3	10.7	7.5 [21]
Calcium (Ca)	10.9	9.9	0.6	1.3	10.8	6.0 [21]
Iron (Fe)	18.6	19.7	23.6	22.7	6.2	20 [21]
Sodium (Na)	4.7	4.8	5.7	7.1	5.7	4.3 [21]
Titanium (Ti)	3.6	4.5	3.6	3.7	2.4	2.3 [21]
Silicon (Si)	6.4	6.2	6.6	6.5	10.6	NA
Trace elements (µg/g)	ASTM D5198					
Potassium (K)	60	100	80	40	1140	420 [21]
Magnesium (Mg)	350	240	30	60	570	460 [21]
Arsenic (As)	102	203	36.2	29.4	28.7	10.7–40.3 [22]
Barium (Ba)	59.1	58.9	47.4	47.4	155.5	124–1380 [22]
Cobalt (Co)	42.5	50.7	3.2	18.9	9.8	5.3–348.9 [23]
Chromium (Cr)	1370	2370	570	640	480	123–1130 [23]
Copper (Cu)	71.2	97.7	11.6	52.0	30.3	31.9–107 [22]
Lithium (Li)	65.1	35.7	5.8	16	287	28.2–162 [22]
Manganese (Mn)	1310	814	73	447	188	NA
Molybdenum (Mo)	4.93	11.6	5.38	6.44	1.46	4.12–13.2 [22]
Nickel (Ni)	123	98.7	24.8	55.3	46.5	59.7–1072 [23]
Lead (Pb)	119	132	43.7	49.4	64.9	47.4–272 [22]
Zinc (Zn)	48	56	17	84	17	33.3–110 [22]
Carbon content (%)	LECO carbon analyzer					
Total carbon	1.3	1.2	0.7	1.1	1.6	NA
Inorganic carbon	1.1	0.9	0.4	0.7	1.0	NA
Organic carbon	0.2	0.3	0.3	0.4	0.6	NA

NA: not available.

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
