# Peer review of "pH-Dependent Leaching Characteristics of Major and Toxic Elements from Red Mud"

_ijerph, 2019, doi:10.3390/ijerph16112046_

Round 1
Reviewer 1 Report
This article has interesting results, discussion about results including figures and tables are not referenced,some of them did not correspond to the one cited in the text, it has to be improve, some mistakes and missing information were founded. More details are included through the article (see the pdf attached).
Author Response
Dear Editor and Reviewers,
Thank you so much for your time and efforts! I, with the other five authors, profoundly appreciate your valuable comments toward the improvement of the paper and your detailed corrections. We have extensively revised our manuscript. Responses to the comments and descriptions of the changes made on the manuscript are given in this file. It should be noted that all page and line numbers in the “REPLY TO REVIEW” refer to the track changes version of the manuscript.
Thanks and best regards,
Jiannan Chen, PhD
Reviewer #1:
COMMENT 1: In lines 23-24, the sentence of “The leaching processes of these elements are independent of the total elemental content in the red mud samples.” doesn´t say clear information, please delete or clarify.
RESPONSE:
Thank you for pointing out this confusion. The revised sentence is presented in lines 21-25.
“At the same pH level, the leaching concentrations of Pb, As, Cr and Cu are generally higher from red mud produced by combined process that those of red mud from Bayer process. The leaching concentration of these elements is not strongly related to the total elemental concentration in the red mud.”
COMMENT 2: It will be better if you can include values mentioned in lines 44-45 related with quality standards of China
RESPONSE:
We apologized for the lacking of the values mentioned in lines 44-45. They had been added in lines 43-51.
“They found that red mud leachate is hyperalkaline (pH>12) and contains high concentrations of aluminum (Al, 118.3–1327.4 mg/L), chloride (Cl−, 511.4–6588.1 mg/L), fluoride (F−, 88.0–299.6 mg/L), sodium (Na, 1200.5–10650.0 mg/L), nitrate (NO3−, 183.2–730.7 mg/L), and sulfate (SO42−, 502.5–6593.0 mg/L). These elements exceed the recommended groundwater quality standards of China up to 6637 times. Sun et al. also found that the minor and trace elements, including arsenic (As, 0.2–2.0 mg/L), chromium (Cr, 0.1–5.9 mg/L), cadmium (Cd, 12–172 μg/L), mercury (Hg, 275–599 μg/L), and selenium (Se, 525–1359 μg/L), have the concentration up to 272 times higher than the maximum contamination levels (MCLs) of groundwater quality standards in both China and the US Environmental Protection Agency (USEPA).”
COMMENT 3: In lines 86 and 234, the discussion about results including figures and tables did not correspond to the one cited in the text.
RESPONSE:
We sincere apologized for the mistake. The inconformity of the figures and tables with cited in test had all been corrected. Please find the revised text in line 101 and 272.
COMMENT 4: I suggest to introduce how the procedure to get the red mud samples and their representative.
RESPONSE:
Thank you for providing this professional comment. The procedure to get the red mud samples and their representative in the text were revised in lines 83-93.
“In this study, red mud samples were collected from five management facilities located in three provinces (i.e., Guangxi, Shandong, and Henan) of China. Red mud samples GX-A-B and GX-B-B were collected from different manufacturers in Pingguo County and in Jingxi County located in Guangxi Province, respectively. Red mud samples SD-A-B and SD-B-B were collected from different manufacturers but in the same area in Zibo, Shandong. Red mud sample HN-A-C was collected in Xingyang, Henan Province. Fresh red mud samples (produced within 7 days), i.e., GX-A-B, GX-B-B, SD-A-B, and HN-A-C, were collected after the pressure filtration (before filled into the red mud reservoir), and the dried red mud SD-B-B was directly sampled from the red mud reservoir. An initial 100 kg of each red mud was collected by a forklift and mixed uniformly by shovels. Then, 20 kg of red mud sample was collected in a sealed container and transported to the laboratory for leaching tests.”
Since the red mud has very fine particles, it was very uniform after mixing.
COMMENT 5: Line 93, I suggest to explain the Method ASTM D2216, ASTM D2487 and ASTM D7348.
RESPONSE:
Thank you for providing the professional comment. Statements have been added to the text.
In lines 101-113: “The moisture content of per red mud was analyzed following the procedure ASTM D2216. 50 g of each red mud was dried in an oven at a temperature of 110±5 °C for 24 h to a constant mass. The moisture content was then calculated based on the masses of water and dry specimen. The moisture content of red mud ranges from 11.0% to 29.0%. The particle size distribution of red mud was tested by ASTM D2487. Each oven-dried (at 110±5 °C) sample was screened by a series of standard sieves, including No. 4 (4.75 mm openings), No. 10 (2 mm), No.14 (1 mm), No. 35 (0.5 mm), No. 60 (0.25 mm), and No. 200 (0.075 mm), respectively. Particle size distribution was calculated by the weight of solid retained on each sieve. Loss on ignition (LOI) was performed by sintering samples at 900 °C using a muffle furnace, and the LOI results of the red mud samples range from 7.7% to 12.8%. According to the Unified Soil Classification System (USCS), red mud samples are mostly sandy or clayey material. GX-B-B is classified as CL-ML (clay-silt with low plasticity), HN-A-C is classified as CH (clay with high plasticity), while GX-A-B, SD-A-B, and SD-B-B are classified as SC (clayey sand).”
COMMENT 6: In Line 115-116, Did you make any experimental design to use these PHs?.
RESPONSE: Thank you for providing this professional comment.
We did not design this experiment. The pH levels were specified by USEPA method 1313. The USEPA method 1313 has been widely used and well recognized to be a reliable method for evaluating the leaching behavior of metals from waste or hazardous material. Studies, such as Chen et al. (2012), and Zhang et al. (2016), have adopted the method to evaluate the release of metals from the cement-based material (red mud has the similar properties). The development of USEPA method 1313 is based on inter-lab results, as well as field-lab comparisons. The corresponding author (J. Chen) has involved in the method development in 2012.
The rationale of each pH is as follows: pH=13±0.5, Upper bound for amphoteric constituents of potential concern (COPCs); pH=12±0.5, Maximum in alkaline range for liquid-solid partitioning (LSP) curves of amphoteric COPCs; pH=10.5±0.5, Substitution if natural pH falls within range of a mandatory target; pH=9±0.5, Minimum of LSP curve for many cationic and amphoteric COPCs; pH=8±0.5, Endpoint pH of carbonated alkaline materials; pH=7±0.5, Neutral pH region; pH=5.5±0.5, Typical lower range of industrial waste landfills; pH=4±0.5, Lower pH limit of typical management scenario; pH=2±0.5, Provides estimates of total or available COPC content.
Chen, J.; Bradshaw, S.; Benson, C.H.; Tinjum, J.M.; Edil, T.B. pH-Dependent Leaching of Trace Elements from Recycled Concrete Aggregate. In GeoCongress 2012; pp. 3729–3738.
Zhang, Y.; Cetin, B.; Likos, W.J.; Edil, T.B. Impacts of pH on leaching potential of elements from MSW incineration fly ash. Fuel 2016, 184, 815–825.
COMMENT 7: Line 175, I suggest to give an explanation which range of value indicate oxidation or reduction?
RESPONSE:
We apologized for the unclear explanation of ORP. The explanation has been added in lines 196-199.
“The oxidation-reduction potential is a widely used parameter for characterizing chemical or biological redox processes. ORP is an indicator of the oxidation or reduction environment (ORP>0: oxidizing environment, ORP<0: reducing environment) [1].”
[1] Lv, Y.; Xiao, K.; Yang, J.; Zhu, Y.; Pei, K.; Yu, W.; Tao, S.; Wang, H.; Liang, S.; Hou, H.; et al. Correlation between oxidation-reduction potential values and sludge dewaterability during pre-oxidation. Water Res. 2019, 155, 96–105.
COMMENT 8: IN Lines 203-206, leaching behavior of Fe and Si, I suggest to explain how this behavior specially the peak observed at pH 10 to 11? In Lines 217-218, I suggest to explain how this behavior for As and Cr at pH 5.5? In Line 236-238, What happened at pH 5.5 with the leaching of F-, explain it.
RESPONSE:
Thank you for providing this thoughtful advice. The statement had been added in the text:
Fe: In lines 234-237: “The solubility of Fe is often controlled by the oxide minerals, e.g., hematite (Fe2O3), which may release Fe at both acidic and alkaline conditions [1]. However, when addition hydroxyl was applied, the Fe tends to precipitate as hydroxide Fe(OH)3 which decreases the Fe concentration in the eluate (pH > 10.5-11).”
Si: In lines 238-240: “In the alkaline condition, Ning et al. [2] also found the Si solubility rise rapidly with pH increasing from 9 to 10.6, where is considered to be the formation of H2SiO42- and H3SiO4- [3].”
Cr and As: In lines 251-255: “Amphoteric elements have relative low released concentration at approximately at pH =5.5~6.5 due to the formation of relatively insoluble hydroxides [4,5]. While, at both high pH and low pH conditions, As and Cr form oxyanion (i.e., AsO43-, HAsO42-, H2AsO4-, H3AsO4, and CrO42-) or cation(i.e. Cr3+), respectively, which are soluble in the eluate [6,7].”
F-: In lines 275-277: “The minimum leaching concentrations occur in neutral and weak acidic conditions (pH=5.5-7). Gong et al. [8] claimed that at weakly acidic pH (5.5-6.5), fluoride is favorable to adsorb on Al2O3.”
[1] Komonweeraket, K.; Benson, C.H.; Edil, T.B.; Bleam, W.F. Leaching behavior and mechanisms controlling the release of elements from soil stabilized with fly ash. Geo-frontiers 2011. Adv. Geotech. Eng. Dallas, Texas, March 13-16, 2011, 1101–1110.
[2] Ning, R.Y.; Tarquin, A.J.; Balliew, J.E. Seawater RO treatment of RO concentrate to extreme silica concentrations. Desalin. water Treat. 2010, 22, 286–291.
[3] Eikenberg, J. On the problem of silica solubility at high pH, 1990.
[4] Malviya, R.; Chaudhary, R. Leaching behavior and immobilization of heavy metals in solidified/stabilized products. J. Hazard. Mater. 2006, 137, 207–217.
[5] Eighmy, T.T.; Eusden, J.D.; Krzanowski, J.E.; Domingo, D.S.; Staempfli, D.; Martin, J.R.; Erickson, P.M. Comprehensive approach toward understanding element speciation and leaching behavior in municipal solid waste incineration electrostatic precipitator ash. Environ. Sci. Technol. 1995, 29, 629–646.
[6] Chen, J.; Bradshaw, S.; Benson, C.H.; Tinjum, J.M.; Edil, T.B. pH-Dependent Leaching of Trace Elements from Recycled Concrete Aggregate. In GeoCongress 2012; pp. 3729–3738.
[7] Cornelis, G.; Johnson, C.A.; Gerven, T. Van; Vandecasteele, C. Leaching mechanisms of oxyanionic metalloid and metal species in alkaline solid wastes: A review. Appl. Geochemistry 2008, 23, 955–976. https://doi.org/10.1016/j.apgeochem.2008.02.001
[8] Gong, W.-X.; Qu, J.-H.; Liu, R.-P.; Lan, H.-C. Adsorption of fluoride onto different types of aluminas. Chem. Eng. J. 2012, 189, 126–133.
COMMENT 9: In Line 315-316 and 352-353, If this study didn´t include any research related with "desorption control mechanism” for As, why you have this conclusion, which is the purpose?
RESPONSE: We apologized the confusion illustration related with “desorption control mechanism” for element As. The discussion about desorption control mechanism for As has been deleted in the text.
Leaching of As may involve arsenate (e.g. Ca, Ba, Pb arsenate) dissolution-precipitation or/and sorption on mineral phases in red mud. However, Visual MINTEQ does not include the database of AsO53-, the further research of As dissolution-precipitation mechanism is in need. The sorption-desorption of As is not within the scope of work of this study.

Reviewer 2 Report
Red mud (bauxite residue) is a highly alkaline and hazardous waste mainly generated from the production of alumina. The leaching behavior of red mud was used to assess the potential risks of red mud leachate to human health ans the environment. The authors chose 6 red muds to detect how the pH value change the leaching process. It is an interesting work. However, the insufficient still need to be revised before publication.
1. Why pH-dependent leaching was focusing on? How to exclude the influence of other factors in the experiment?
2. Authors should fully introduce how pH influence elements leachate in the introduction, and the mechanisms on leachate process and results. Even other factors influencing on element leachate should also be summarized and compared with pH. However, only one sentence from 62-63 is not enough.
3. The authors can group results and discussion together if it is necessary because some discussion content were found in the results.
4. Furthermore, I don’t think the discussion is wonderful, and more conditions should be considered, while not only compared the same or different from other publications. For example, (1) leachate influenced by pH could be changed when the temperature changing, (2) one element could be also be influenced by other ions.
Author Response
Dear Editor and Reviewers,
Thank you so much for your time and efforts! I, with the other five authors, profoundly appreciate your valuable comments toward the improvement of the paper and your detailed corrections. We have extensively revised our manuscript. Responses to the comments and descriptions of the changes made on the manuscript are given in this file. It should be noted that all page and line numbers in the “REPLY TO REVIEW” refer to the track changes version of the manuscript.
Thanks and best regards,
Jiannan Chen, PhD
Reviewer #2:
COMMENT 1: Please, attention I suggest an explain why pH-dependent leaching was focusing on? How to exclude the influence of other factors in the experiment?
RESPONSE:
Thank you for providing this professional comment.
In many respects leaching behavior is reflected by the pH dependence and this analysis may provide an understanding of the environmental impact. Furthermore, pH is shown to be the most important governing parameter controlling the leaching characteristics of elements.
In this study, the other factors, like temperature was at atmospheric temperature, L/S (10/1) and reaction time (24h, according to the particle size) were fixed by USEPA Method 1313 recommends, with a constant speed of 30rpm agitated in an end-over-end tumbler. The L/S (10/1) is used because this L/S ratio has been widely used, and the results in the current study can be compared to results from other studies. Preliminary tests on the effect of contact time (0~72 h) on the leaching experiment indicated 24 h is sufficient for each batch to reach chemical equilibrium condition (pH, EC, and elements reached constant).
COMMENT 2: I suggest to fully introduce how pH influence elements leachate in the introduction, and the mechanisms on leachate process and results. Even other factors influencing on element leachate should also be summarized and compared with pH.
RESPONSE:
Thank you for providing thoughtful advice on the pH and other factors influencing on element leachate.
Statements have been added to the text:
In lines 63-74: “The leaching of elements, especially hazardous elements, is largely dependent on the environmental conditions. These conditions include pH, temperature, reaction time, liquid to solid ratio. Uzun et al. [1] evaluated the leaching of metal from red mud by increasing agitation rate, they found the total dissolution increased from 5% to 23.18% by agitating up to 400 rpm. Rachel et al. [2] found acid addition (5mol/L) and temperature (80 °C) can significantly enhance the metal extraction from red mud. Lim and Shon [3] found that acid concentration (6N sulfuric acid), leaching temperature (70 °C), and reaction time (2 hours with ultrasound) could enhance the metal leaching from red mud, while the increase of solid to liquid ratio (from 2% to 4%) reduces the metal leaching. Among these factors, pH is the most influential parameter that controls the release of inorganic constituents from the solid phase [4,5]. The leaching behavior as a function of pH will help estimate the mobility of elements from red mud in various environmental conditions of geotechnical applications.”
In lines 220-221: “Cama et al. [6] found aluminosilicate phases are less stable compared to boehmite and gibbsite under acid attack.”
In lines 221-224: “The concentration of Ca shows a continuous negative correlation with pH, thus representing the cationic pattern (Fig. 3b). The decrease in pH will induce strong acid attack on the Ca-bearing minerals, thus release a higher concentration of Ca in the eluate [7].”
In lines 234-237: “The solubility of Fe is often controlled by the oxide minerals, e.g., hematite (Fe2O3), which may release Fe at both acidic and alkaline conditions [8]. However, when addition hydroxyl was applied, the Fe tends to precipitate as hydroxide Fe(OH)3 which decreases the Fe concentration in the eluate (pH > 10.5-11).”
In lines 237-240: “The leaching of Si is majorly due to the dissolution of silicates rather than quartz under acid attack [9]. In the alkaline condition, Ning et al. [10] also found the Si solubility rise rapidly with pH increasing from 9 to 10.6, where is considered to be the formation of H2SiO42- and H3SiO4- [11].”
In lines 251-255: “Amphoteric elements have relative low released concentration at approximately at pH =5.5~6.5 due to the formation of relatively insoluble hydroxides [12,13]. While, at both high pH and low pH conditions, As and Cr form oxyanion (i.e., AsO43-, HAsO42-, H2AsO4-, H3AsO4, and CrO42-) or cation(i.e. Cr3+), respectively, which are soluble in the eluate [14,15].”
In lines 262-263 and 266-268: “ Previous studies indicated that the leaching of Ba and Mn are sensitive to the pH of the environment [16,17].” “Like Ca, leaching of Mg and Ba tend to increase with decreasing pH due to competition with the hydrogen ion. Additionally, Astrup et al. [18] found that leaching of Ba in eluate is associated with solubility of Ba(S, Cr)O4.”
In lines 268-270: “At pH=2, the maximum release was observed (1.3-16.7 mg/L) for Mn, and the concentration decreases with increasing pH until pH=8, where Mn2+ cation tends to precipitate as Mn(OH)2 [19].”
In lines 275-277: “The minimum leaching concentrations occur in neutral and weak acidic conditions (pH=5.5-7). Gong et al. [20] claimed that at weakly acidic pH (5.5-6.5), fluoride is favorable to adsorb on Al2O3.”
[1] Uzun, D.; Gülfen, M. Dissolution kinetics of iron and aluminium from red mud in sulphuric acid solution. Indian J. Chem. Technol. 2007.
[2] Pepper, R.A.; Couperthwaite, S.J.; Millar, G.J. Comprehensive examination of acid leaching behaviour of mineral phases from red mud: Recovery of Fe, Al, Ti, and Si. Miner. Eng. 2016, 99, 8–18.
[3] Lim, K.; Shon, B. Metal components (Fe, Al, and Ti) recovery from red mud by sulfuric acid leaching assisted with ultrasonic waves. Int. J. Emerg. Technol. Adv. Eng 2015, 5, 25–32.
[4] Mudd, G.M.; Weaver, T.R.; Kodikara, J. Environmental geochemistry of leachate from leached brown coal ash. J. Environ. Eng. 2004, 130, 1514–1526.
[5] Fruchter, J.S.; Rai, D.; Zachara, J.M. Identification of solubility-controlling solid phases in a large fly ash field lysimeter. Environ. Sci. Technol. 1990, 24, 1173–1179.
[6] Cama, J.; Metz, V.; Ganor, J. The effect of pH and temperature on kaolinite dissolution rate under acidic conditions. Geochim. Cosmochim. Acta 2002, 66, 3913–3926.
[7] Tiruta-Barna, L.; Imyim, A.; Barna, R. Long-term prediction of the leaching behavior of pollutants from solidified wastes. Adv. Environ. Res. 2004, 8, 697–711.
[8] Komonweeraket, K.; Benson, C.H.; Edil, T.B.; Bleam, W.F. Leaching behavior and mechanisms controlling the release of elements from soil stabilized with fly ash. Geo-frontiers 2011. Adv. Geotech. Eng. Dallas, Texas, March 13-16, 2011, 1101–1110.
[9] Knauss, K.G.; Wolery, T.J. The dissolution kinetics of quartz as a function of pH and time at 70 °C . Geochim. Cosmochim. Acta 1988, 52, 43–53.
[10] Ning, R.Y.; Tarquin, A.J.; Balliew, J.E. Seawater RO treatment of RO concentrate to extreme silica concentrations. Desalin. water Treat. 2010, 22, 286–291.
[11] Eikenberg, J. On the problem of silica solubility at high pH, 1990.
[12] Malviya, R.; Chaudhary, R. Leaching behavior and immobilization of heavy metals in solidified/stabilized products. J. Hazard. Mater. 2006, 137, 207–217.
[13] Eighmy, T.T.; Eusden, J.D.; Krzanowski, J.E.; Domingo, D.S.; Staempfli, D.; Martin, J.R.; Erickson, P.M. Comprehensive approach toward understanding element speciation and leaching behavior in municipal solid waste incineration electrostatic precipitator ash. Environ. Sci. Technol. 1995, 29, 629–646.
[14] Chen, J.; Bradshaw, S.; Benson, C.H.; Tinjum, J.M.; Edil, T.B. pH-Dependent Leaching of Trace Elements from Recycled Concrete Aggregate. In GeoCongress 2012; pp. 3729–3738.
[15] Cornelis, G.; Johnson, C.A.; Gerven, T. Van; Vandecasteele, C. Leaching mechanisms of oxyanionic metalloid and metal species in alkaline solid wastes: A review. Appl. Geochemistry 2008, 23, 955–976.
[16] Sauer, J.J.; Benson, C.H.; Aydilek, A.H.; Edil, T.B. Trace elements leaching from organic soils stabilized with high carbon fly ash. J. Geotech. Geoenvironmental Eng. 2011, 138, 968–980.
[17] Bin-Shafique, M.S.; Benson, C.H.; Edil, T.B. Leaching of heavy metals from fly ash stabilized soils used in highway pavements. Geo Engineering Program, Department of Civil and Environmental Engineering, 2002.
[18] Astrup, T.; Dijkstra, J.J.; Comans, R.N.J.; der Sloot, H.A.; Christensen, T.H. Geochemical modeling of leaching from MSWI air-pollution-control residues. Environ. Sci. Technol. 2006, 40, 3551–3557.
[19] Gitari, W.M.; Fatoba, O.O.; Petrik, L.F.; Vadapalli, V.R. Leaching characteristics of selected South African fly ashes: effect of pH on the release of major and trace species. J Env. Sci Heal. A Tox Hazard Subst Env. Eng 2009, 44, 206–220.
[20] Gong, W.-X.; Qu, J.-H.; Liu, R.-P.; Lan, H.-C. Adsorption of fluoride onto different types of aluminas. Chem. Eng. J. 2012, 189, 126–133.
COMMENT 3: I suggest to group results and discussion together if it is necessary because some discussion contents were found in the results.
RESPONSE:
We apologized for the repetition contents between discussion and results.
We have group the necessary discussion with the results, but still keep a section of “Discussion” to highlight the research findings. However, the repetitive content has been removed and necessary supplementary sentences had been added in the text.
In lines 348-351: “Zhang et al. [1] stated the leaching mechanism of Ca from MSWI fly ash was controlled by calcium sulfates (gypsum (CaSO4.2H2O) and anhydrite (CaSO4)), while in more alkaline condition (pH>12), calcium carbonates (calcite (CaCO3) played a more important role, as well as this study.”
In lines 354-357: “Murarka et al. [2] and Dijkstra et al. [3] also found that tenorite (CuO) and malachite (Cu2(OH)2CO3) controlled Cu leaching in coal-combustion residues and MSW bottom ash tests, which are same controlled minerals in the red mud.”
In lines 362-364: “Mudd et al. [4] and Fruchter et al. [5] claimed that barite (BaSO4) and carbonate compounds (i.e., witherite (BaCO3)) are likely formed in aqueous solutions of fly ashes, which could also be the controlling mineral for Ba from red mud.”
[1] Zhang, Y.; Cetin, B.; Likos, W.J.; Edil, T.B. Impacts of pH on leaching potential of elements from MSW incineration fly ash. Fuel 2016, 184, 815–825.
[2] Murarka, I.P.; Rai, D.; Ainsworth, C.C. Geochemical basis for predicting leaching of inorganic constituents from coal-combustion residues. In Waste Testing and Quality Assurance: Third Volume; ASTM International, 1991.
[3] Dijkstra, J.J.; Meeussen, J.C.L.; der Sloot, H.A. Van; Comans, R.N.J. A consistent geochemical modelling approach for the leaching and reactive transport of major and trace elements in MSWI bottom ash. Appl. Geochemistry 2008, 23, 1544–1562.
[4] Mudd, G.M.; Weaver, T.R.; Kodikara, J. Environmental geochemistry of leachate from leached brown coal ash. J. Environ. Eng. 2004, 130, 1514–1526.
[5] Fruchter, J.S.; Rai, D.; Zachara, J.M. Identification of solubility-controlling solid phases in a large fly ash field lysimeter. Environ. Sci. Technol. 1990, 24, 1173–1179.
COMMENT 4: I suggest to consider more conditions in the discussion while not only compared the same or different from other publications. For example: (1) leachate influenced by p could be changed when temperature changing, (2) one element could be also be influenced by other ions.
RESPONSE:
Thank you for providing this professional comment.
In this paper, we tend to research the influence of pH on leaching elements from red mud. Other factors like leaching temperature, L/S ratio, agitation speed are fixed. In the future, we will do more experimental studies about changing other factors.
However, we discussed some interact of elements in the text.
In lines 267-268: “Additionally, Astrup et al. [1] believed Ba leaching in eluate was associated with solubility of Ba(S, Cr)O4 solid solution as well.”
In lines 275-277: “The minimum leaching concentrations occur in neutral and weak acidic conditions (pH=5.5-7). Gong et al. [2] claimed that at weak acidic pH (5.5-6.5), fluoride is favorable to adsorb on Al2O3.”
[1] Astrup, T.; Dijkstra, J.J.; Comans, R.N.J.; der Sloot, H.A.; Christensen, T.H. Geochemical modeling of leaching from MSWI air-pollution-control residues. Environ. Sci. Technol. 2006, 40, 3551–3557.
[2] Gong, W.-X.; Qu, J.-H.; Liu, R.-P.; Lan, H.-C. Adsorption of fluoride onto different types of aluminas. Chem. Eng. J. 2012, 189, 126–133.

Round 2
Reviewer 2 Report
accept